# Higher Risk of Recurrence in Patients Treated for Head and Neck Cancer with Low BMI and Elevated Levels of C-Reactive Protein

**DOI:** 10.3390/cancers14205161

**Published:** 2022-10-21

**Authors:** Diana Spiegelberg, Christer Malmberg, Ylva Tiblom Ehrsson, Göran Laurell

**Affiliations:** 1Department of Surgical Sciences, Uppsala University, 75185 Uppsala, Sweden; 2Department of Immunology, Genetics and Pathology, Uppsala University, 75185 Uppsala, Sweden; 3Department of Medical Sciences, Uppsala University, 75185 Uppsala, Sweden

**Keywords:** CRP, thrombocytes, BMI, leukocytes, squamous cell carcinoma, relapse

## Abstract

**Simple Summary:**

Head and neck cancer (HNC) treatment poses several challenges in clinical practice, and treatment side effects can be debilitating due to the close proximity of important anatomical structures. Cancer recurrence post-treatment presents some of the most challenging HNC management issues. This prospective study identifies high-risk groups for recurrence of head and neck cancer, based on commonly accessible clinical parameters. In this study with 272 HNC patients, elevated pre- and post-treatment CRP levels, low BMI and advanced stage at admission indicate higher risk for recurrence of disease. Using these parameters, a risk model is proposed which may be useful for estimating the probability of cancer recurrence and allow the identification of high and low-risk patients.

**Abstract:**

This prospective study identifies high-risk groups for recurrence of head and neck cancer by BMI and circulating inflammatory response markers. Head and neck cancer patients from three Swedish hospitals were included (*n* = 272). Leukocyte and thrombocyte counts, CRP levels, and BMI were measured pre-treatment and post-treatment. Associations between the four factors and treatment failure (residual tumor, loco-regional failure, general failure/distant metastasis) were assessed using a Cox proportional hazards model adjusted for sex, age at the initial visit, smoking status, cancer stage, and hemoglobin count. CRP level was the only significant single variable, with an average increase in risk of recurrence of 74% (*p* = 0.018) for every doubling. The predictive power of a combined model using all variables was highest during the initial months after treatment, with AUC under the ROC curve 0.75 at the 0–3 month timepoints. Patients with elevated pre- and post-treatment CRP levels are at higher risk for recurrence of disease. Male patients with low post-treatment BMI, advanced stage, and high CRP at any time post treatment are at high risk for recurrence. The combined model may be useful for stratifying post-treatment patients into low and high-risk groups, to enable more detailed follow-up or additional treatment regimens.

## 1. Introduction

Head and neck cancer (HNC) remains one of the top 10 most common causes of cancer-related deaths in Europe and North America, and mortality is high worldwide [1]. HNCs are heterogeneous in nature and arise in the mucosal linings of the upper aerodigestive tract, with alcohol consumption and smoking identified as strong risk factors. Over the past 30 years, there has been a steady decrease in the number of carcinomas related to smoking and excessive alcohol consumption; however, the role of human papillomavirus (HPV) as a cause has emerged, particularly in oropharyngeal cancer [2]. In general, HPV-positive oropharyngeal cancer in non-smokers is more responsive to the available treatment options and has therefore a significantly greater overall 5-year survival rate compared to HPV-negative HNC [3]. Even though HPV is a major independent prognostic factor of head and neck squamous cell carcinoma (HNSCC), there is a broad variety of treatment responders and non-responders within patient groups, and additional markers are needed to identify high and low-risk groups at an early stage of the disease.

There is some evidence that an increase in systemic inflammatory responses is associated with cancer incidence [4,5] and recurrence, for example in urological, lung, and colorectal cancer, and furthermore that this increase, in turn, is responsible for the decrease in disease-free and overall survival [5,6,7,8]. C-reactive protein (CRP) is one of the most important markers for systemic inflammatory response and is routinely used as a diagnostic marker in the clinical setting. CRP is produced in the liver and responds to infections and inflammations due to the release of cytokines such as interleukin-1 (IL-1), interleukin-6 (IL-6), and tumor necrosis factor alpha (TNF-α). CRP has long been considered exclusively to be a marker for cardiovascular disease and infectious disease; however, it may function as a critical marker for the development and progression of malignancies as well [9]. Furthermore, the baseline thrombocyte count (Trc) and leukocyte count (Lkc) are routinely investigated in the clinic. Thrombocytes, also called platelets, are the smallest circulating blood cells and ensure the integrity of blood vessels and are involved in the clotting process. Several studies have shown that thrombocytes can promote tumor cell proliferation by releasing growth factors, chemokines, proangiogenic regulatory proteins, and proteolytic enzymes [10,11]. Both low and high thrombocyte levels have been implicated in cancer. Recent research suggests that thrombocytosis is a promising general marker for early stage cancer, especially for patients with lung or colorectal cancer [11]. Similarly, a high Lkc can indicate inflammatory processes caused by a number of different medical conditions, e.g., cancer treatment, infection, stress, trauma, or allergy [12].

Access to a basic method of differentiating patient groups with low or high risk for recurrence would be of high clinical value because recurrence (local, regional, or distant) is associated with high risk for mortality. The earlier that recurrence is detected, the earlier that therapy can be re-initiated or adjusted to salvage therapy, thus probably increasing the likelihood of survival. The systemic inflammatory response might be useful as a predictive marker in HNC, for example, by measuring the CRP level and Lkc in patient full blood as an easy and cost-effective approach, especially because these laboratory readings are routinely performed at patient visits and are therefore readily available. However, CRP and Lkc are acute-phase markers, typically rising at 6–12 h after infection for CRP and 5–24 h for Lkc, and then dropping when the inflammation abates. However, chronic inflammation associated with cancer may elevate these markers long-term. Therefore, repeated measurements of these factors could potentially be more informative in predicting treatment response and disease-free and overall survival. For example, there exists some support for the notion that patients with HNC with increased baseline CRP values have a worse prognosis [13,14], however, these processes are not fully understood in cases of HNC.

In addition, body weight or body mass index (BMI) was suggested to be a predictive marker for cancer patient outcomes in a study where patients with high BMI had lower mortality rates compared to patients with normal and low BMI [15]. This phenomenon is also referred to as the “obesity paradox” because usually a low BMI is associated with a healthy lifestyle thus reducing the risk of developing cardiovascular disease or cancer [16]. To our knowledge, the details of how changes in BMI affect risk for recurrence in contrast to high or low initial BMI have not been explored.

As mentioned, the measurement of body weight, CRP, and thrombocyte and lymphocyte status is routinely implemented in daily clinical practice. However, to our knowledge no prospective observational study has yet been conducted analyzing trends in repeated measurements of BMI, CRP, Lkc, and Trc in patients with HNC. The aim of this study was therefore to investigate whether the levels of these relatively inexpensive standard blood analysis factors in patients with HNC can be related to the risk of a residual tumor within 6 months or with recurrence of the disease (>6 months after treatment). Lkc, Trc, and CRP values and initial BMI and BMI change were correlated with other independent risk factors such as clinical stage at diagnosis, age, smoking habits, and anemia pre-treatment and at several timepoints up to 24 months post-treatment.

## 2. Materials and Methods

### 2.1. Study Subjects

For this prospective observational study, a mixed cohort of 272 patients with HNC was diagnosed, treated and followed up at three HNC centers in Sweden—namely, Uppsala University Hospital (Uppsala), Örebro University Hospital (Örebro), and the University Hospital of Umeå (Umeå)—from November 2015 to August 2020. Of these, 198 patients were male (73%) and 74 female (27%), and the mean age was 63 years (range 32 to 89 years) at baseline. Written informed consent was obtained from all subjects included in this investigation according to Swedish legislation. Inclusion criteria included curable, newly diagnosed untreated HNC with a 0–2 WHO performance status. Only patients treated with curable intent were included. Distributions of the anatomic site of the cancer, stage of disease, p16/HPV status and treatments are presented in Table 1. The most common primary tumor sites included p16/HPV positive oropharynx (*n* = 117), p16/HPV negative oropharynx (*n* = 7), oral cavity (*n* = 78) and larynx (n = 31). Patients were treated according to Swedish national guidelines and treatment options were discussed at multidisciplinary meetings. Patients were divided into four treatment groups and altogether 242 patients underwent radiotherapy, either as single modality treatment or combined modality treatment (surgery, concomitant chemotherapy/target therapy). More detailed information on treatment groups, the distribution of treatment and treatment modalities for patients with oropharynx cancer, oral cavity cancer and larynx cancer can be found in Table 1 and Appendix A. p16 immunohistochemistry (n = 48) or PCR for HPV DNA detection (n = 76) were used to establish HPV tumor status in patients with oropharyngeal cancer. Exclusion criteria included prior treatment of malignant tumors within the last 5 years (with the exception of skin cancer), immune suppressant treatment, severe alcohol problems, cognitive impairments, or other inability to participate or inability to understand Swedish. The study was approved by the Regional Ethical Review Board in Uppsala (2014/447) and registered at ClinicalTrials.gov, NCT03343236. Tumor response to treatment and loco-regional status were regularly assessed by ENT physicians. The standard follow-up after radiotherapy (with or without concomitant chemotherapy or cetuximab) for oropharyngeal cancer was PET/CT three months after the termination of treatment.

All patients underwent dietary monitoring according to local guidelines and, if indicated, received complementary nutritional supplement treatment. A study representative met with the patients before treatment (pre-treatment), 7 weeks after treatment start (post-treatment), and at 3, 6, 12, and 24 months after treatment termination. Body weight was measured and blood was drawn from the patients at four occasions (before treatment, 7 weeks after start of treatment, and at 3 and 12 months after termination of treatment). BMI was calculated as body weight divided by the square of height (kg/m^2^), as measured using a weight scale and stadiometer without outdoor clothing or shoes.

### 2.2. Data Collection, Demographics, and Disease-Specific Data

All data were prospectively collected from the patients and from their medical records. The data were collected from the first clinical visit up to 24 months. This study used the general Swedish follow-up plan for head and neck cancer patients, and recurrence events were tracked up to 30 months after the initial visit. Blood diagnostics were carried out in certified laboratories at the local hospital (CRP, Lkc, Trc, and hemoglobin (Hb)) and were available at pre-treatment, post-treatment, and at 3 months and 12 months after the termination of treatment. Each patient was followed up by repeated clinical examinations, and findings of recurrence of tumor were documented. No uniform definition of recurrence exists [17,18,19]. In the present study, we have used 6 months as a cutoff and defined recurrence as follows: Recurrence was divided into residual disease (defined as loco-regional recurrence <6 months post-treatment), loco-regional recurrence (defined as loco-regional recurrence ≥6 months post-treatment), or general failure/distant metastasis (defined as distant metastasis >6 months post-treatment).

### 2.3. Statistical Analysis

Associations between the covariates of CRP, Lkc, Trc, BMI at first visit, BMI change (baselined at post-treatment), and the outcome of disease recurrence were assessed using a Cox proportional hazards model. After the initial visit (pre-treatment baseline), follow up started at the post-treatment timepoint (7 weeks after start of treatment) and was terminated either at the date of documented recurrence, date of death, or date of censoring. The censoring date for an individual was set to the next planned visit following the latest recorded follow-up visit. Covariates were updated at each visit (post-treatment, 3 months, and 12 months) and values were assumed to be valid until the next visit. Recurrence events were tracked up to 30 months after the start of treatment. The model was adjusted for age at the initial visit, sex, smoking status, cancer stage, Hb counts, and pre-treatment CRP, Trc, and Lkc levels by including these factors as covariates. Due to the high number of patients with p16/HPV positive oropharyngeal cancer and lack of data in other tumor locations, HPV status was not included as a covariate.

Knowing that the model might overfit the data due to the relatively high number of degrees of freedom per case, a post-analysis variable selection method was used. This selection step used the multivariate and non-linear Boruta algorithm, which employs randomized variables in an iterative random forest survival classifier. The Boruta method was performed with 100 steps, using the ranger random forest implementation and Z-scores of mean decrease in accuracy as an importance measure. Any variable less important than the best-performing random variable was excluded and a reduced Cox regression model was used for validation and prediction [20]. Variables which were borderline significant were examined further by estimated partial dependence plots from a random forest classifier based on the Boruta variables [21], and excluded or kept based on the relationship with recurrence. HNSCC site and treatment type was also tested at this stage, since they conceivably could be important. Possible violation of the proportional hazards assumption of the Cox regression was assessed using plots of the scaled Schoenfeld residuals against time. The discriminative ability of the model was assessed by a time-dependent area under the ROC curve (AUC), as well as ROC curves at 3 months, 1 and 2 years respectively. The Efron-Gong optimism bootstrap was used to correct the AUC values at each failure time for overfitting [22]. Briefly, the model was re-fitted in bootstrap samples and the AUC values were calculated both on the bootstrap data and on the original data. The difference between the AUC values when comparing the bootstrap data and the original data is a measure of optimism in the apparent AUC values. The optimism-corrected AUC values were obtained by averaging 2000 bootstrap estimates of the optimism and subtracting those averages from the apparent AUC values. Further details of this approach are found elsewhere [23]. The reduced Cox proportional hazards model with only the informative variables from the full model, as well as a model with only CRP as a factor, was used for cross-validation and prediction using the same approach as above. All analyses were performed using R version 3.5.0 [24] with functions from the survival [25], Epi [26], Boruta [20], randomForestSRC [21] and risksetROC [27] packages.

## 3. Results

### 3.1. Patient Data

The distributions of the variables used in the Cox model are summarized in Table 2. Recurrence rates varied across the time points, from 4.4% (post-treatment to 3 months) to 16.2% (3 months to 12 months). The mean age decreased over the course of the study, reflecting an expected higher mortality in older patients [28]. Smoking status was registered only at diagnosis, and the change in ratio between smokers/non-smokers during the study was due to mortality. The distribution of smoking status was similar in the different groups over time. As for the stage of disease, the most common stage at diagnosis was stage 1 (39.9%).

### 3.2. Patterns of Treatment Failure

A total of 71 patients developed recurrence of disease within the study period. Forty patients were diagnosed with recurrence of disease within 6 months after treatment and were thus classed as residual disease, whereas 31 patients were diagnosed with recurrence after 6 months. These were grouped by type of recurrence into loco-regional failure or general failure/distant metastasis (Table 2). Thirty-six patients receiving chemotherapy did not tolerate all planned weekly doses of cisplatin, 12 of these patients displayed a recurrence. Moreover, a dose reduction of radiotherapy was observed in 3 patients. None of these displayed a recurrence.

### 3.3. Association of CRP, BMI, Hb, Trc, and Lkc with Recurrence

During the study, it was observed that median CRP levels were markedly increased in the recurrence group compared to the non-recurrence group (defined as patients where recurrence occurred or did not occur, respectively, after the measurement but before the next follow-up visit) following treatment, and at post-treatment follow-up the median CRP levels were 34 mg/L compared to 18.7 mg/L in the recurrence group and non-recurrence group, respectively (Table 3). CRP levels increased in both the recurrence and non-recurrence groups from pre-treatment baseline to post-treatment. From the time of diagnosis onward, the mean Lkc, Trc, and Hb values did not differ significantly between the recurrence and non-recurrence groups. BMI changes were also similar in both patient groups.

### 3.4. Cox Proportional Hazards Model of Outcome Predictors

To further investigate the impact of each variable, a Cox proportional hazards model was used with adjustments for the independent risk factors of smoking, age, clinical stage, and hemoglobin counts as well as for the initial values (pre-treatment). The significance of each factor/variable and the proportional hazards over the change in the measured variables can be seen in Figure 1 and Table 4. It is evident that elevated CRP level after the start of treatment was strongly correlated with risk for overall recurrence of disease (74% increased risk per doubled CRP value, *p* = 0.018). Elevated Trcs were weakly correlated with risk for recurrence of disease, with a moderate effect size (34% increase) but lower significance (*p* = 0.138). There was no evidence of Lkcs being correlated with either increased or decreased risk for recurrence. For CRP, even a moderately elevated value from baseline led to high risk, but the additional increased risk at very high CRP levels was low. For Trc, the risk increased almost linearly with measured Trc, but, as mentioned, this relationship was not statistically significant. Divided by recurrence type, it is clear from Table 4. that high CRP levels were mainly predictive of residual disease (*p* = 0.004), with an effect size of 119%, and not loco-regional recurrence (*p* = 0.763). Loco-regional recurrence was not significantly correlated with increased CRP, and no subgroup of recurrence was significantly correlated with Trc or Lkc.

There was no discernible relationship between BMI loss and disease recurrence. However, BMI at post-treatment was strongly correlated with recurrence, where high initial BMI was beneficial. The effect size was a 46% reduction in risk for recurrence per BMI unit over the mean BMI of 25.5 (*p* = 0.005). The independent risk factors of sex and cancer stage at diagnosis were significantly correlated with recurrence, while age and smoking status were not. The individual contribution of each factor to the fit of the Cox regression model can be seen in Figure 1a, as analyzed by the Wald test.

The empirical cumulative distribution functions for the three blood markers can be seen in Figure 2 divided into groups by whether recurrence occurred before the next visit or not. It is evident that of the three examined blood markers, the CRP marker had the highest correlation with the recurrence of disease before the next visit compared to the other two markers. The largest effect for CRP was post-treatment. It was also observed that Trc and Lkc counts were almost identical between the recurrence group and non-recurrence group at all time points.

In total, the predictive power of the current model was good, and the AUC value (Figure 3a) varied between 0.7 and 0.75 during the first year post-treatment with a peak at 3 months after treatment start. Figure 3b) also includes the time-varying AUC for a model with only CRP. The AUC for the model containing all variables was consistently higher than the AUC for the model with only CRP, thus indicating the necessity of using all variables for the prediction. However, when split by a subgroup of recurrence, it is clear that CRP alone was still a good prognostic marker for residual disease, whereas for loco-regional relapse and general recurrence the performance was close to random. For the recurrence subgroups, it is clear that the main predictive power of the full model was for residual disease, where the main peak in AUC up to >0.75 was seen. For loco-regional recurrence and general relapse, the AUC value was stable at 0.65 irrespective of time (Figure 3d). This agrees well with the hazard ratios for the subgroups, which indicate that CRP and stage are the main significant predictors for residual disease, with baseline BMI, sex, and stage significant for loco-regional recurrence and only stage significant for general recurrence (Table 4).

To investigate the ability of this model to stratify high and low-risk populations, a reduced Cox regression model with only the important variables from the full model was used, as determined by Boruta variable selection (Trc, CRP, pre-treatment BMI, and clinical stage, details in Appendix A). The time-varying AUC for the reduced Cox model was not significantly different from the full model (Figure 3c). This model was used to predict the rates of recurrence for the mean population (mean CRP, mean pre-treatment BMI, mean Trc, as well as the mean of the other variables), as well as a high-risk group (CRP > 90th percentile at any time, initial BMI < 20, but mean for the other variables), for each of the clinical stages at diagnosis (Figure 4). For clinical stages II-IV, the high-risk population had a greater than 50% probability of recurrence over a 24-month period post-treatment, while the mean population had less than ~25% probability of recurrence. Since stage was such a significant factor, an attempt was made to perform subgroup analysis using the reduced Cox model. However, the recurrence count was too low in stages I-III to yield informative data (Appendix A). For further model validation, the correlation to the clinically important parameters treatment type or HNSCC site were included in the Boruta variable selection, but they were not significantly more important than the random shadow variables in this dataset.

Furthermore, the risk of recurrence over 2 person-years as a function of initial BMI, CRP at any time and clinical stage at diagnosis was calculated using the reduced Cox regression hazard rates (Figure 5). This risk model could potentially be used for risk estimation of patients with HNC undergoing treatment, by entering BMI and clinical stage at diagnosis, combined with continuous follow up of CRP. However, the predictive accuracy of this risk model remains to be evaluated.

## 4. Discussion

Despite recent advances in the diagnosis and treatment of HNC over the last few decades, there has been only moderate improvement in 5-year survival rates. It is clear that increasing treatment success and survival rates will require better diagnostics to allow for earlier start of first-line therapy and optimization of the current treatment regimens as well as earlier and accurate diagnostics of recurrence, which may help to better select patients for salvage therapy. Individualized therapy is likely to provide strong benefits, especially considering the heterogenous nature of HNCs, but this will require new methods for patient stratification.

To our knowledge, this prospective study represents the first attempt to provide a predictive model for HNSCC recurrence based on commonly registered patient statistics such as systemic inflammatory response markers from whole blood, as well as BMI measurements.

In a retrospective study, an association between neutrophil-to-lymphocyte ratio, CRP levels, BMI and overall survival as well as recurrence in HNSCC has been previously reported [29]. In addition, a high BMI seems to indicate a better prognosis compared to low BMI groups [15,30]. A previous study by our research group showed that overweight or obese patients (BMI > 25) with oropharyngeal cancer at the start of radiotherapy had better overall survival compared to patients with low BMI, indicating that BMI at the start of radiotherapy can be used as a prognostic factor for 5-year overall survival [15]. However, weight loss per se was not a negative prognostic factor in these two studies. Moreover, it seemed that patients with higher BMI could cope better with body weight loss and did not relapse to the same extent [15]. This trend was also evident in the present study, where low pretreatment BMI was strongly correlated with an increased risk of recurrence. As an example from other studies, BMI was recently evaluated as a prognostic marker in glioblastoma multiforme, demonstrating that patients with an elevated BMI had significantly better overall survival. However, the mechanism of this interaction is not fully understood and needs further investigation [31].

In this study, we found a significant association between elevated levels of the inflammatory marker CRP and recurrence of disease, especially recurrence of disease <6 months, which we defined as residual tumor.

The association with recurrence >6 months post-treatment is less clear, but it is important to note that the covariates in this study were only measured at post-treatment, 3 months, and 12 months after the termination of treatment. Because the correlation between CRP and recurrence before the next measurement was most pronounced at post-treatment and 3 months, and for recurrence up to 6 months post-treatment, it is possible that the large gap between the 3 months and 12 months sampling points mask transient increases in CRP for patients with recurrences after the 6-months checkpoint.

A follow-up study to further clarify the connection between long-term but transient CRP increases is warranted based on these results.

There are several possible mechanistic explanations for the described correlation between inflammatory response, low BMI, and recurrence of disease. In general, two types of inflammation can be distinguished. During acute inflammation, which lasts for a short time, edema and migration of leukocytes is induced. The second form is systemic inflammation, which is characterized by an increase in lymphocytes and macrophages as well as the proliferation of blood vessels and connective tissue [32,33]. An increase in CRP is usually considered an acute-phase marker [9], but this study suggests a correlation between long-term increasing trends and recurrence of disease. This circulating inflammatory response marker typically rises within the same day of an infection, after which a drop occurs when the inflammation abates. In cancer, however, long-term inflammation has been suggested to induce a sustained low-level increase in inflammatory markers. For thrombocytes, studies suggest a relationship between thrombocytosis and risk for later cancer diagnosis, thus likely indicating their utility as an early phase diagnostic marker [11,34]. Our study shows a weak correlation between elevated Trc and risk for recurrence of disease, but this was not significant at the 95% level. Even though this association was not significant, the effect size was large and could be indicative of an actual finding that should be followed up with a larger population.

Even though the data presented here support earlier findings that a high BMI is associated with better prognosis in HNC patients, it must be pointed out that the obesity paradox has been questioned by several authors [35,36]. Interestingly, obesity can influence the production of circulating inflammatory mediators, including the production of the cytokine IL-6 that plays a role in the development and progression of various diseases [37,38]. IL-6 stimulates CRP production, fibrinogen production, leukocyte release, endothelium activation, and hemostasis [37,39], which seems contradictive to the results of this study. However, the increase in inflammatory markers caused by obesity might be smaller compared to the increase caused by the systemic inflammation due to carcinoma. Clearly, a complex relationship exists between tumors, adipose tissue and the immune system, and further studies are needed to assess if excessive body fatness per se has favorable effects during treatment of HNC.

This prospective study clearly indicates the potential clinical utility of using repeated measurements of CRP and possibly Trc to predict recurrence of disease in HNC patients. These measurements are particularly attractive because they can be made objectively and inexpensively in clinical practice worldwide. CRP and Trc may be useful as indicators of disease progression and for monitoring disease progression. Especially because the effect size of elevated CRP was so large (estimated at 12–171% increased risk for recurrence per doubling of CRP at any time during follow-up), an elevated CRP value in HNC patients at 7 weeks after treatment start to 6 months post-treatment should be a strong signal for follow-up investigations aiming to detect potential recurrence of disease. The finding is supported by a recent review presenting a meta-analysis based on 17 studies pointing out that pretreatment elevated CRP indicates a poor prognosis in patients with HNC [40]. Management of recurrence is complex, and treatment decisions are often influenced by a complex mix of factors. There is accumulating evidence that surgery should be reserved for selected patients where the site of primary tumor and recurrence-free interval are important factors to consider [40], and thus identifying factors that might influence the success of treatments is important. One recent study showed that asymptomatic recurrence may be a positive prognostic factor for salvage treatment [41], and selecting this high-risk subpopulation might allow the use of more involved and resource-demanding diagnostic methods, which can be difficult to implement for all patients. Therefore, stratification of patients into low and high recurrence risk groups based on CRP might be a viable strategy. This study quantified the increased risk of recurrence coupled to elevated inflammatory markers. Because a diagnosis of recurrence can be assumed to always occur a significant time after the recurrence happens on a cellular level, a detected increase in CRP should likely be seen as a signal of ongoing local or metastatic regrowth. Furthermore, our data suggest that a combination of CRP and initial BMI, together with clinical stage and sex, might have value as an algorithm for stratifying patient populations into a high-risk group (>50% probability of recurrence over 2 years of follow up, BMI < 20 and CRP > 30 at any time during follow-up). Similarly, these data could be used for identifying low-risk patients (high BMI, low CRP) with low probability of recurrence. The risk model presented in Figure 5 can potentially be used for stratifying patient populations based on which risk level is acceptable, however the accuracy of the model needs to be further evaluated with additional prospective data. One possible uncertainty is CRP expression depending on other covariates in the study, for example stage or treatment type. For example, a recent study by Astradsson et al. demonstrated differential expression patterns of inflammatory proteins and immune response markers between patients that underwent surgery, radiotherapy and/or chemoradiation. This study showed that cisplatin-based chemoradiotherapy had immunological effects in HNSCC patients [42], whereas surgery alone did not. In our current study, we observed a significantly lower CRP level at the post-treatment follow-up in patients receiving surgery as only treatment. Since patients receiving surgery only are over-represented in early clinical stages in our data, and stage was strongly correlated with recurrence, there is a risk of confounding in the connection between CRP and recurrence. Even so, this study adds to the evidence of the link between the presence of a systemic inflammatory response and cancer recurrence. It is possible that further research in this area may lead to novel treatment strategies.

To implement the presented results, we suggest that patients with high levels of CRP and platelets and/or low BMI pre-treatment should be examined at shorter intervals and with more detailed follow-up visits. It has been discussed elsewhere that the most important goal for post-treatment follow up is early detection of recurrence, which has a positive impact on prognosis due to better outcomes of salvage therapy. However, early detection is rarely possible (only up to 20%), and there is a lack of prognostic tumor markers for recurrence in HNC [43,44]. At the same time, routine diagnostic imaging for detecting recurrence is impractical and costly. As such, CRP monitoring over time such as presented here might be a useful way to delineate a high-risk group of patients where more common, expensive, and thorough follow up visits can be motivated. At the same time, a reduced treatment plan for low-risk patients may be beneficial to avoid over-treatment and reduce unnecessary and unpleasant side effects. However, the association between CRP and thrombocyte levels as well as BMI need to be explored further in order to fully understand the association with the long-term goal of implementing these findings into clinical practice and increasing the overall survival of patients with HNC.

## 5. Conclusions

Our data indicate that head and neck cancer patients with elevated pre- and post-treatment CRP levels are at higher risk for recurrence of disease, especially residual disease. Furthermore, cancer stage and BMI at initial visit were found to be significantly important predictors for recurrence. The combined risk model proposed in this work might be useful for stratifying post-treatment patients into low and high-risk groups, which could have important clinical implications for the post treatment management of these patients.

## Figures and Tables

**Figure 1 cancers-14-05161-f001:**
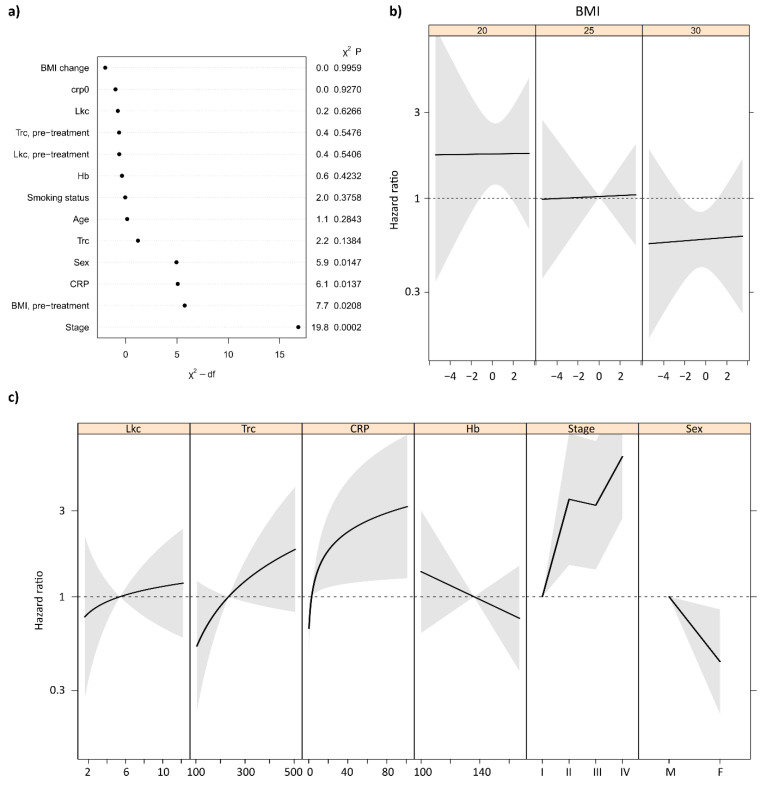
(**a**) Individual contribution to the model fit from each factor in the full Cox regression model, as tested by the Wald test. (**b**) Hazard ratios of BMI change for different categories of initial BMI. No significant differences were seen. A hazard ratio of 1 indicates no change from baseline. (**c**) Hazard ratios for the examined factors in the adjusted Cox proportional hazards model relative to the median value for each factor. For CRP there was a significant correlation between high CRP counts and increased risk of recurrence, while for Trc, Hb, and Lkc the correlation was not significant.

**Figure 2 cancers-14-05161-f002:**
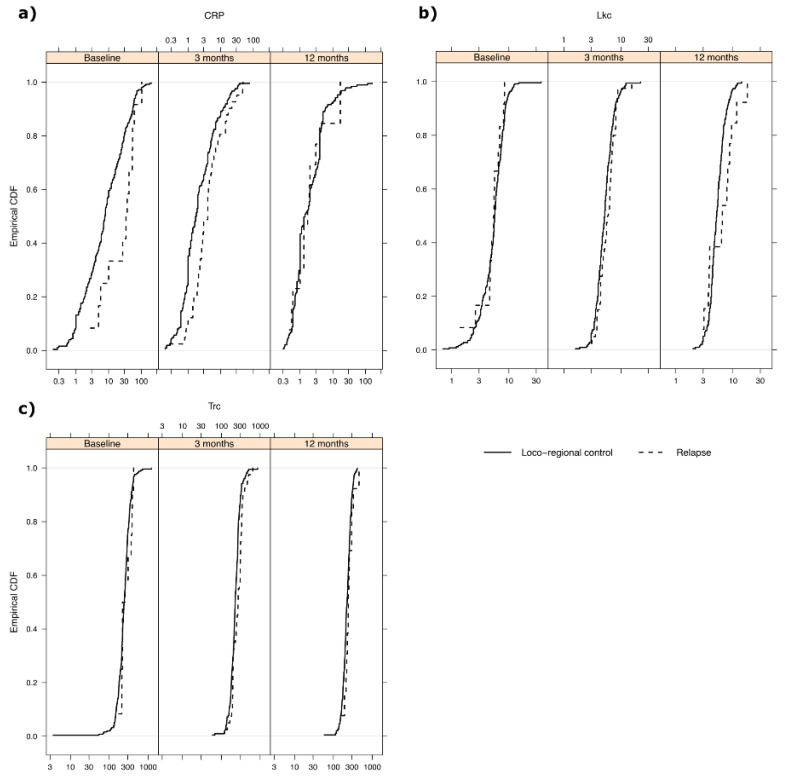
Empirical cumulative distribution functions for the three examined prognostic blood markers. (**a**) CRP was elevated in the recurrence group, defined as recurrence occurring before the next follow-up. CRP elevation in this group was evident at pre-treatment and at 7 weeks and 3 months after the start of treatment. (**b**) Lkc counts did not differ between recurrence and loco-regional control groups at any time point. (**c**) Trc counts did not differ between recurrence and loco-regional control groups at any time point.

**Figure 3 cancers-14-05161-f003:**
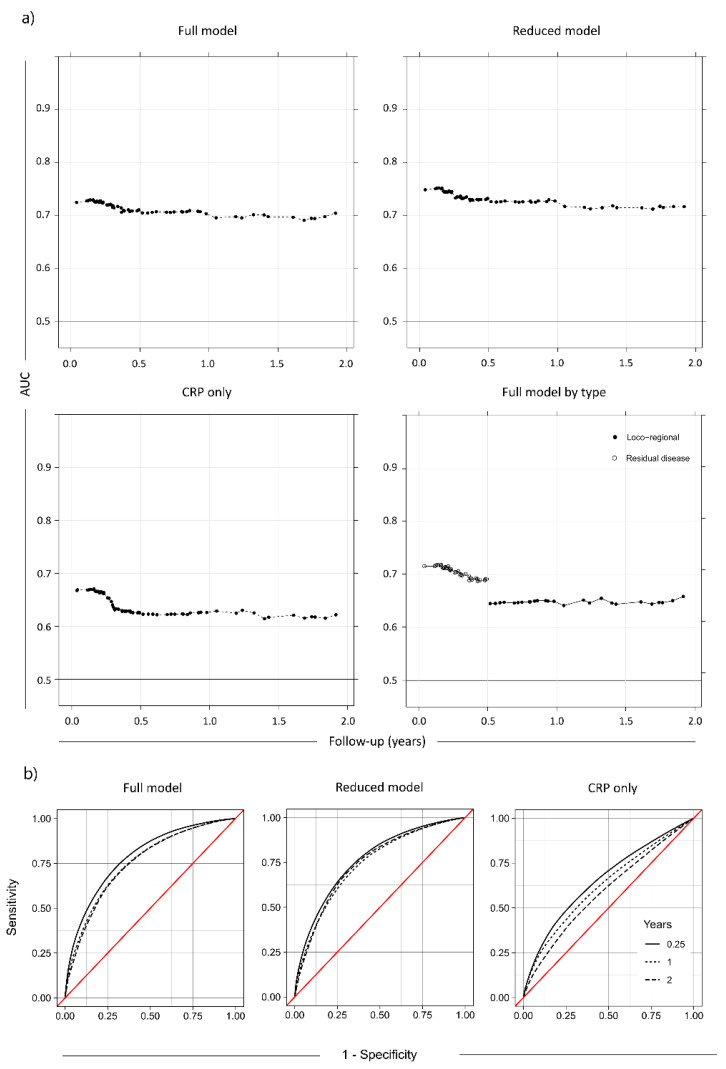
(**a**) Full model: AUC for Cox proportional hazards model over time for all variables. The AUC value (the area under the ROC curve) is an estimate of the predictive power of the model, where high AUC indicates low false positive and/or false negative rates and AUC = 0.5 is random chance. The AUC showed a clear peak at 0–6 months. Reduced model: AUC over time for a reduced model only using the significant factors (Trc, CRP, pretreatment BMI, and stage). The AUC was not appreciably different from the full model. CRP only: AUC over time for only CRP. It is evident that CRP alone was a worse predictor than the full model. Full model by type: The full model divided by recurrence type. The model was most predictive of residual disease, followed by loco-regional recurrence and general relapse. (**b**) ROC curves for the models in a). The ROC curves are calculated at 3 months, 1 year and 2 years of follow-up time.

**Figure 4 cancers-14-05161-f004:**
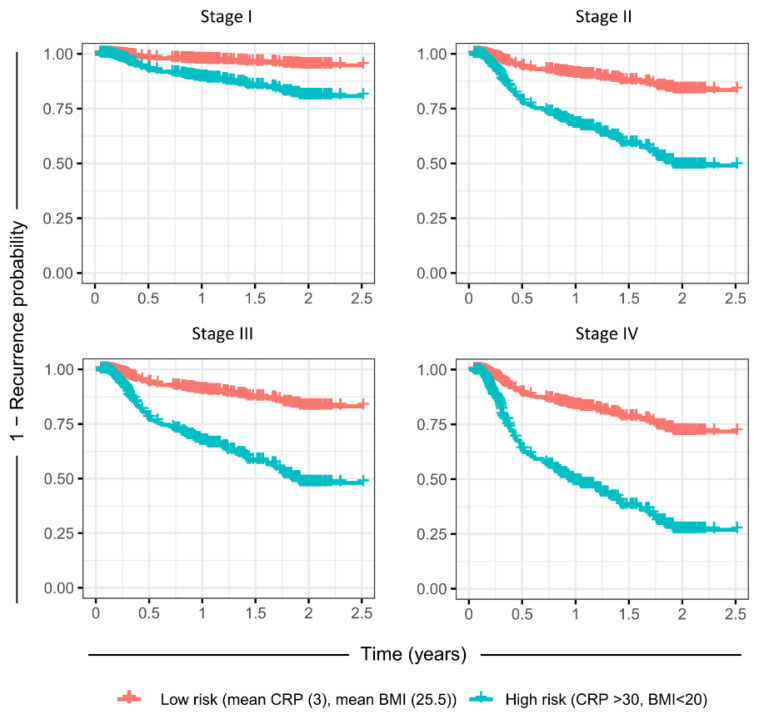
Predicted recurrence probability from the reduced Cox regression model with only significant factors included. Red indicates the mean of the study population (mean CRP, mean initial BMI) and blue indicates the higher risk population (CRP > 30, initial BMI < 20). The sex factor is set at average risk for males/females.

**Figure 5 cancers-14-05161-f005:**
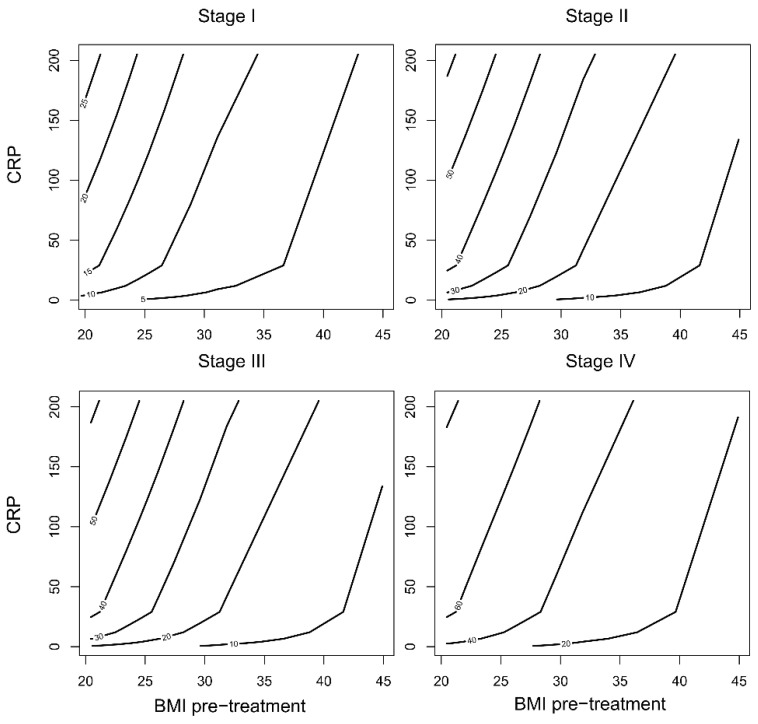
Risk estimation charts. The dataset was simultaneously partitioned by deciles of initial BMI and CRP, yielding 100 cells. In each cell the risk of recurrence over 2 person-years was calculated from the reduced Cox regression model hazard rates, for each of the stages at diagnosis. The contours delineate the risk of recurrence depending on the initial BMI and CRP measured at any timepoint, in percent.

**Table 1 cancers-14-05161-t001:** Anatomic site and stage of the disease at pre-treatment and treatment of the disease.

Anatomic Site	N	Stage I	Stage II	Stage III	Stage IV	Treatment	N
Oropharynx	124	68	21	31	4	Surgery only	30
Oral cavity	78	18	20	12	28	Radiotherapy **	150
Larynx	31	14	6	6	5	Chemoradiation ***	75
Hypopharynx	6	0	0	1	5	Radiotherapy with	
Nasopharynx	8	1	3	1	3	Cetuximab ****	17
Unknown primary	9	4	0	4	1		
Salivary gland	8	1	2	1	4		
Nose and sinus	4	3	1	0	0		
Other	4 *	1	0	1	0		

* Stage not known for all cases, ** All patients who received radiotherapy, with or without surgery, *** All patients who received radiotherapy with concomitant cisplatin, with or without surgery, **** All patients who received radiotherapy with concomitant cetuximab, with or without surgery.

**Table 2 cancers-14-05161-t002:** Patient descriptives, covariates, and outcomes used in the Cox proportional hazards model. The (x/y) in the table are the 25/75 percentile values or are N (%) for the categorical variables. The model combines the four different levels.

Covariates	Initial Visit (Pre-Treatment Baseline)	Post-Treatment	3 Months	12 Months	24 Months
Total (N)	276	272	259	211	154
BMI	26.7 (23.4–29.3)	25.5 (22.5–28.0)	25.1 (22.2–27.7)	25.6 (22.3–28.4)	26.1 (23.0–28.4)
Sex ratio (M/F)	2.6	2.7	2.8	2.6	2.6
CRP	7.9 (1.1–8.2)	19.8 (2.6–26.0)	5.1 (1.0–5.0)	4.7 (0.8–4.0)	-
Trc	274.4 (228.0–311.2)	258.7 (199.0–300.0)	245.1 (194.0–273.5)	230.9 (193.0–265.0)	-
Lkc	7.5 (5.9–8.8)	5.9 (4.2–7.3)	5.6 (4.2–6.6)	5.6 (4.2–6.5)	-
Hb	139.8 (131.1–149.0)	130.0 (119.2–139.0)	136.4 (128.0–146.0)	141.0 (133.0–149.0)	-
BMI change	-	0	−0.5 (−1.2–0.3)	−0.3 (−1.2–0.95)	-0.1 (-0.8–1.3)
Age	63.1 (56.0–71.0)	63 (56.0–71.0)	63.0 (56.0–71.0)	62.6 (55.5–70.5)	63.1 (56.3–71.0)
Smoking status:					
Never	94 (34.1)	91 (33.5)	86 (33.2)	70 (33.2)	50 (32.5)
Former	157 (56.9)	156 (57.4)	149 (57.5)	123 (58.3)	91 (59.1)
Current	25 (9.1)	25 (9.2)	24 (9.3)	18 (8.5)	13 (8.4)
Stage:					
I	110 (39.9)	110 (41.6)	108 (41.9)	103 (48.8)	81 (52.6)
II	53 (19.2)	53 (18.7)	53 (20.5)	43 (20.4)	27 (17.5)
III	58 (21.0)	57 (18.2)	53 (20.5)	39 (18.5)	25 (16.2)
IV	53 (19.2)	50 (17.3)	44 (17.1)	26 (12.3)	21 (13.6)
Unknown	2 (0.7)	2 (0.7)	1 (0.4)	0 (0)	0 (0)
**Outcome Variables ***	**Initial Visit–Post Treatment**	**Post-Treatment–3 Months**	**3 Months–12 Months**	**12 Months–24 Months**	**>24 Months**
Recurrence, total (N)	-NA	12	42	15	2
Residual disease within 6 months (N)	-	12	28	0	0
Recurrence of disease after 6 months (N)	-	0	14	15	2
Loco-regional failure (N)	-	10	28	9	2
General failure (N)	-	2	14	6	0

* Recurrence N: recurrence which occurred after indicated visit but before next visit. Recurrence was divided into residual disease (defined as loco-regional recurrence <6 months post-treatment), loco-regional recurrence (defined as loco-regional recurrence >6 months post-treatment) or general failure/distant metastasis (defined as distant metastasis >6 months post-treatment).

**Table 3 cancers-14-05161-t003:** Median differences in covariates between the recurrence and non-recurrence groups (where each group is defined by recurrence occurring or not recurring before the next follow-up visit).

Variable	Pre-Treatment	Post-Treatment		3 Months		12 Months		24 Months	
	*N = 276*	*Non-recurrence group,* *N = 259*	*Recurrence group,* *N = 12*	*Non-recurrence group,* *N = 210*	*Recurrence group,* *N = 42*	*Non-recurrence group,* *N = 195*	*Recurrence group,* *N = 15*	*Non-recurrence group,* *N = 151*	*Recurrence group,* *N = 2*
CRP	7.9 (1.1–8.2)	18.7 (2.4–24.0)	36.2 (8.9–53.3)	4.2 (1.0–4.6)	7.7 (2.0–7.0)	3.9 (0.8–4.0)	4.0 (1.0–3.0)	-	-
Trc	274 (228–311)	258 (197–296)	218 (211–218)	238 (191–268)	238 (201–325)	229 (193–264)	260 (221–295)	-	-
Lkc	7.5 (5.9–8.8)	5.9 (4.2–7.3)	5.4 (4.7–6.6)	5.2 (4.2–6.5)	6.1 (4.4–7.0)	5.5 (4.3–6.4)	7.2 (3.8–8.7)	-	-
Hb	139.8 (131–149)	131 (120–139)	130 (122–141)	138 (128–146)	132 (125–141)	141 (134–150)	141 (124–149)	-	-
BMI loss	-	0	0	−0.5 (−1.2–0.3)	−0.3 (−1.2–0.3)	−0.3 (−1.2–0.9)	0.3 (−0.4–1.2)	0.1 (−0.75–1.35)	−1.25 (−2.1-0.42)

**Table 4 cancers-14-05161-t004:** Hazard ratios for selected variables and factors in the Cox regression model.

Variable	Hazard Ratio			
	Recurrence, Total (f0)	Residual Disease (f2)	Loco-Regional Recurrence (f)	General Recurrence (f1)
CRP ^1^	1.74 (1.12–2.71), *p* = 0.018 *	2.19 (1.29–3.73), *p* = 0.004 *	0.87 (0.35–2.15), *p* = 0.763	0.62 (0.12–3.11), *p* = 0.559
Trc ^1^	1.34 (0.90–1.99), *p* = 0.138	1.38 (0.88–2.18), *p* = 0.163	1.38 (0.68–2.78), *p* = 0.372	1.99 (0.62–6.40), *p* = 0.250
Lkc ^1^	1.10 (0.74–1.65), *p* = 0.627	1.04 (0.65–1.67), *p* = 0.872	1.49 (0.76–2.91), *p* = 0.247	1.13 (0.39–3.26), *p* = 0.823
Hb ^1^	0.83 (0.52–1.31), *p* = 0.423	0.85 (0.48–1.53), *p* = 0.594	0.76 (0.36–1.63), *p* = 0.485	0.99 (0.29–3.34), *p* = 0.983
BMI change ^2^	1.00 (0.86–1.18), *p* = 0.970	1.00 (0.79–1.28), *p* = 0.909	1.00 (0.76–1.33), *p* = 0.887	1.24 (0.84–1.84), *p* = 0.967
BMI, 7 weeks ^2^	0.54 (0.35–0.83), *p* = 0.005 *	0.57 (0.32–1.02), *p* = 0.058	0.44 (0.22–0.91), *p* = 0.027 *	0.48 (0.14–1.74), *p* = 0.267
Age ^2^	0.82 (0.57–1.18), *p* = 0.284	0.87 (0.56–1.36), *p* = 0.548	0.76 (0.39–1.47), *p* = 0.417	0.75 (0.26–2.14), *p* = 0.589
Sex (m:f)	0.44 (0.22–0.84), *p* = 0.015 *	0.51 (0.23–1.17), *p* = 0.115	0.28 (0.08–0.93), *p* = 0.038 *	0.79 (0.15–4.12), *p* = 0.778
Smoker (y:n)	1.30 (0.72–2.33), *p* = 0.383	1.35 (0.64–2.83), *p* = 0.429	1.34 (0.51–3.53), *p* = 0.551	0.91 (0.19–4.36), *p* = 0.916
Stage (I:IV)	6.00 (2.72–13.3), *p* < 0.001 *	5.86 (2.13–16.1), *p* < 0.001 *	8.09 (2.15–30.3), *p* = 0.002 *	9.88 (1.44–67.6), *p* = 0.020 *

^1^ Per doubling, ^2^ Per unit increase, * Significant factor.

## Data Availability

Data is maintained in this article. Data is not publicly available due to privacy.

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
