# Peer review of "Higher Risk of Recurrence in Patients Treated for Head and Neck Cancer with Low BMI and Elevated Levels of C-Reactive Protein"

_cancers, 2022, doi:10.3390/cancers14205161_

Round 1

Reviewer 1 Report

Spiegelberg and etc. assessed the predictive value of two clinical indicators (BMI and C-reactive protein) in the HNC prognosis. BMI and C-reactive protein were two general clinical indicators without invasive and un-expensive. Interestingly, it was differentiating from previous perception that HNC patients with higher pre-treatment BMI had a better progression. This study was provided a new perception of BMI in cancers. Some major commons should be responded before publish.

Major comments

1.     In the study cohort, the number of males was much higher than that of female, it might cause a statistical bias. The author should explain whether this quantity variance caused an unauthentic differential progression between the male and female.

2.     The author should provide more data to explain why Trc and LkC were chosen to assess the progressions in HNC, neutrophil and lymphocyte were more associated to inflammation of patients.

3.     Missing patient information: e.g. alcohol and EGFR status

4.     Rationale for recurrence classification, 6 months for cutoff, literature support?

5.     Rationale for time point setting for post-treatment data collection?

6.     A single indicator is not sufficient to evaluate patients’ inflammatory or nutritional status, and there is measurement error in BMI. In contrast, the Glasgow Prognostic Score may be a better indicator to study, or others such as NLR, PLR and LDH, which are all easily available in routine blood tests.

7.     Subgroup analysis is necessary, especially for stage and treatment modalities.

8.     For Figure 3, a Time_ROC or Time_AUC plot is required, which can be combined to make it more visible.

Author Response

Spiegelberg and etc. assessed the predictive value of two clinical indicators (BMI and C-reactive protein) in the HNC prognosis. BMI and C-reactive protein were two general clinical indicators without invasive and un-expensive. Interestingly, it was differentiating from previous perception that HNC patients with higher pre-treatment BMI had a better progression. This study was provided a new perception of BMI in cancers. Some major commons should be responded before publish.

Answer: We thank the reviewer for this straightforward summary of the manuscript.

  1. In the study cohort, the number of males was much higher than that of female, it might cause a statistical bias. The author should explain whether this quantity variance caused an unauthentic differential progression between the male and female.

Answer: We agree with the reviewer that this is a problematic feature with our dataset. However, this is not unexpected in our cohort. Approximately two thirds of the Swedish patients with head and neck cancer are men. We have added a variable selection stage in the statistical analysis where we evaluate all variables with the powerful multidimensional and non-linear Boruta variable selection algorithm. Here, we see that indeed sex is not associated with recurrence, and the association in the Cox regression model is likely due to the skewed dataset. We have updated the text and reduced Cox model accordingly.

  1. The author should provide more data to explain why Trc and LkC were chosen to assess the progressions in HNC, neutrophil and lymphocyte were more associated to inflammation of patients

Answer: We agree that investigating neutrophils and lymphocytes could have been a better choice. However, we have chosen to use blood samples (CRP, Lkc, Trc, and Hb) that are routinely taken in a clinical context in Sweden, and neutrophil and lymphocyte are not measured routinely here.

  1. Missing patient information: e.g. alcohol and EGFR status

Answer: We thank the reviewer for the suggestion to add these data, but with regards to EGFR status please see our answer above Q2. In Sweden, EGFR status was relevant for patients with head and neck cancer who were considered for treatment with cetuximab (Erbitux). However, cetuximab therapy is only used very rarely and the EGFR status is not measured routinely.

Regarding the alcohol use, the patients have been asked if they have severe alcohol problems with cognitive impairments making them unavailable to participate in the study. We agree that more information of alcohol use would be interesting. On the other hand, it is well-known that many patients do not admit that they have drinking problems. Thus, this is a classical error in clinical statistics. Furthermore, severe alcoholism is relatively rare in Sweden.

Thank you again for pointing this out. Regarding exclusion criteria we have now added “severe alcohol problems with cognitive impairments”. Line 157-158.

  1. Rationale for recurrence classification, 6 months for cutoff, literature support?

Answer: We would like to thank the reviewer for this valuable comment. We know that no uniform definition of locally recurrent or loco-regional recurrence exist. We have added to the manuscript (line 194-196). “No uniform definition of recurrence exists (17-19). In the present study we have used 6 months as a cutoff and defined recurrence as follows: Recurrence was divided into residual disease (defined as loco-regional recurrence <6 months post-treatment).”

New cited references:
17. Gorphe P, Moya-Plana A, Guerlain J, Tao Y, Nguyen F, Breuskin I, et al. Disease-free time stratification in locally recurrent head and neck carcinoma after definitive radiotherapy or chemoradiotherapy. Eur Arch Otorhinolaryngol. 2022;279(6):3063-9.

  1. Miller JA, Moradi F, Sundaram V, Liang R, Zhang C, Nguyen NK, et al. Posttreatment FDG-PET/CT Hopkins criteria predict locoregional recurrence after definitive radiotherapy for oropharyngeal squamous cell carcinoma. Head & neck. 2022.
  2. McSpadden R, Zender C, Eskander A. AHNS series: Do you know your guidelines? Guideline recommendations for recurrent and persistent head and neck cancer after primary treatment. Head & neck. 2019;41(1):7-15.

  1. Rationale for time point setting for post-treatment data collection?

Answer: We agree with the reviewer that the rationale was unclear. As this is an observational study where no attempt is made to affect the outcome we have used the general Swedish follow plan for head and neck cancer patients. We have now added this rationale to the manuscript, line 187-189. “This study used the general Swedish follow-up plan for head and neck cancer patients, and recurrence events were tracked up to 30 months after the initial visit.”

  1. A single indicator is not sufficient to evaluate patients’ inflammatory or nutritional status, and there is measurement error in BMI. In contrast, the Glasgow Prognostic Score may be a better indicator to study, or others such as NLR, PLR and LDH, which are all easily available in routine blood tests.

Answer: We agree with the reviewer that there are other parameters for measuring nutritional status, which may be superior the ones we used. However, in the present study we were limited to the clinical praxis in Sweden, i.e. BMI. We have added a text clarifying the method used to measure BMI (line 174-176). “BMI was calculated as body weight divided by the square of height (kg/m2), as measured using a weight scale and stadiometer without outdoor clothing or shoes.”

Previously, Valdes at al. assessed the mentioned inflammatory and nutritional status in HNSCC (Glasgow Prognostic Score, NLR etc.). We have added this study in the discussion (line 451-453), “In a retrospective study an association between neutrophil-to-lymphocyte ratio, CRP levels, BMI and overall survival as well as recurrence in HNSCC has been previously reported (30).”

New cited reference: 

30. Valdes M, Villeda J, Mithoowani H, Pitre T, Chasen M. Inflammatory markers as prognostic factors of recurrence in advanced-stage squamous cell carcinoma of the head and neck. Curr Oncol. 2020;27(3):135-41.

  1. Subgroup analysis is necessary, especially for stage and treatment modalities.

Answer: We thank the reviewer for this suggestion, but we have to disagree with regards to method. Subgroup analysis would within the framework of our study likely be problematic due to the low case count compared to the number of variables involved. We did track different treatment modalities, as evident from Table 1. However (instead of subgroup analysis), we added the Boruta variable selection step, which is robust to low case counts and high dimensionality in the dataset. The Boruta statistic did not assign any relevance to treatment type (Figure S1a), but Stage was very strongly associated with recurrence. As for stage subgrouping, we attempted to perform the subgroup analysis and included the results in the manuscript, Table S2 and page 10, line 389-396. However, subgrouped by stage the low recurrence count in each group meant that no reliable data was gained, even from the reduced Cox model. We added in the manuscript: “Since stage was such a significant factor, an attempt was made to perform subgroup analysis using the reduced Cox model. However, the recurrence count was too low in stages I-III to yield informative data (Figure S2). For further model validation, the correlation to the clinically important parameters treatment type or HNSCC site were included in the Boruta variable selection, but they were not significantly more important than the random shadow variables in this dataset.”

  1. For Figure 3, a Time_ROC or Time_AUC plot is required, which can be combined to make it more visible.

Answer: We thank the reviewer for the suggestion and have now added Figure 3b, ROC curves at 3 timepoints for each model (3 months, 1 year and 2 years).

Reviewer 2 Report

Comments and suggestions for authors

·         Although interesting, the findings of this study are not new and have been published previously by other authors, decreasing the originality and novelty of the current paper

·         One of the clinical end points of this study is local recurrence, which is defined as a recurrent tumor within 3cm of the primary lesion and within 3 years (= 39 months) of diagnosis of the primary lesion (Rhode et al Eur Arch Otorhinolaryngol. 2020). However, recurrence events in the current study are tracked up to 30 months, and patients are included up until August 2020, meaning that these last patients do not even have 25 months of follow-up. Why didn’t you track recurrence events until 39 months?

·         Why was alcoholism an exclusion criterion? It is a major causative agent of HNSCC and probably also a predictor or residual/recurrent disease and associated with BMI.

·         There is no information about the treatment intent of the included patients. Were all patients treated with curative intent? Or were there also patients with palliative regimens? This information seems very relevant to me.

·         There is also no information on whether patients did or did not complete their treatment regimen. This is relevant information too, since patients who did not complete their treatment regimens are at increased risk of residual or recurrent disease.

·         Why was HNSCC site not included as covariate? Prognosis of stage I laryngeal cancer is completely different from stage I hypopharyngeal cancer. How was the distribution of recurrences among the different HNSCC sites? I estimate hardly any recurrences in the HPV-positive OPSCC group, which is 43% of your study population.

·         Why was treatment not included as covariate?

·         Why was smoking status only registered at diagnosis? Was there any information on whether patients continued smoking during/after treatment or if they stopped? It is known that continuing smoking during radiotherapy significantly decreases treatment effect due to tissue hypoxia from smoking (Chen et al Int J Radiat Oncol Biol Phys 2011).

·         From the 272 patients included, 117 (43%!) are HPV-positive OPSCC. HPV is known to be the strongest prognostic factor in HNSCC, yet you decided not to include them as covariate “due to the high number of patients with p16/HPV positive oropharyngeal cancer and lack of data in other tumor locations”. I do not understand this. It is generally known that HPV-status does not affect prognosis in other HNSCC sites, so you could have made two groups (HPV-positive OPSCC versus HPV-negative HNSCC). Or you could have excluded HPV-positive OPSCC at all. But you cannot exclude them as covariate when it’s prognostic impact is so strong.

·         In Table 2 it is stated that BMI was on average 26.7 at baseline and 25.5 post-treatment, yet the BMI change posttreatment is zero, how is that possible?

·         Did you check whether post-treatment CRP was correlated to therapy? The post-treatment timepoint was set 7 weeks after start of treatment, but I would expect that in patients that only required surgery, CRP is lower 7 weeks postoperative than in patients requiring a radiotherapy regimen of 7 weeks, in which the last RT fraction was given right before the post-treatment timepoint.

·         You state that “patients with elevated pre- and post-treatment CRP levels are at higher risk for recurrence of disease”. But did you check why these patients had elevated CRP levels? Did they have comorbidities? Active infections? Immune-suppressant medications? Any contra-indications to receive optimal treatment? Were they fit enough to finish the whole treatment regimen?

·         And while you state that elevated CRP is associated with higher risk for recurrence, later on in the paper it seems to be true only for residual disease, and not for loco-regional recurrence.

·         You state that “BMI post-treatment was strongly correlated with recurrence”, but are patients with a low BMI not also at increased risk of not being able to complete their treatment? Did you correct for this? Did these patients follow-up on the advices the dietician gave them? I think the Cox proportional hazards model currently includes too few covariates to make this statement.

Author Response

  1. Although interesting, the findings of this study are not new and have been published previously by other authors, decreasing the originality and novelty of the current paper

Answer: We thank the reviewer for this frank assessment of the originality of the study. To our knowledge however, the current study represents the first attempt at a predictive model for HNSCC recurrence based on the simple and easily accessible parameters BMI and Crp. The association between Crp, BMI and recurrence in HNSCC has been described before (ref), in an aetiological study. Our manuscript both presents an aetiological study using Cox regression, but also a predictive model from the reduced Cox regression. Furthermore, this manuscript has a larger sample size. We have added (Valdes M. et al. Inflammatory markers as prognostic factors of recurrence in advanced-stage squamous cell carcinoma of the head and neck. Curr Oncol. 2020 Jun;27(3):135-141) as a reference and line 447-453 in the discussion. “To our knowledge, this prospective study represents the first attempt to provide a predictive model for HNSCC recurrence based on commonly registered patient statistics such as systemic inflammatory response markers from whole blood, as well as BMI measurements.

In a retrospective study an association between neutrophil-to-lymphocyte ratio, CRP levels, BMI and overall survival aa well as recurrence in HNSCC has been previously reported (30). In addition, a high BMI seem to indicative a better prognosis compared to low BMI groups (31, 32).”

  1. One of the clinical end points of this study is local recurrence, which is defined as a recurrent tumor within 3cm of the primary lesion and within 3 years (= 39 months) of diagnosis of the primary lesion (Rhode et al Eur Arch Otorhinolaryngol. 2020). However, recurrence events in the current study are tracked up to 30 months, and patients are included up until August 2020, meaning that these last patients do not even have 25 months of follow-up. Why didn’t you track recurrence events until 39 months?

Answer: We thank the reviewer for reading the manuscript thoroughly and giving us valuable feedback. However, this study is an observational study, different from a defined clinical trial. The data collection in this manuscript is still ongoing. Due to the scarcity of HNSCC patients, this overall study may continue for many years. The Cox-regression model we used takes into account right-censored data, and it is especially suited for this data type, where patients can be censored at any timepoint. Once the study concludes, we will revisit these questions with the final dataset.

  1. Why was alcoholism an exclusion criterion? It is a major causative agent of HNSCC and probably also a predictor or residual/recurrent disease and associated with BMI.

Answer: We thank the reviewer for pointing this out, the exclusion criterion is now supplemented with a correct text change in manuscript: regarding exclusion criteria we have added “severe alcohol problems”. Line 157-158. In Sweden relatively few patients are affected by excessive alcohol use. Furthermore, these patients are in general not willing to enter a study and sign an informed consent document.

  1. There is no information about the treatment intent of the included patients. Were all patients treated with curative intent? Or were there also patients with palliative regimens? This information seems very relevant to me

Answer: We disagree with the reviewer, as it is stated on line 139: Inclusion criteria included curable, newly diagnosed untreated HNC with a 0–2 WHO performance status. I. e. all patients included were treated with curative intent. To avoid confusion, we have clarified this on line 140-141. “Only patients treated with curable intent were included.”

  1. There is also no information on whether patients did or did not complete their treatment regimen. This is relevant information too, since patients who did not complete their treatment regimens are at increased risk of residual or recurrent disease.

Answer: We thank the reviewer for this suggestion, and have now added the relevant information on page 6, line 273-277. “Thirty-six patients receiving chemotherapy did not tolerate all planned weekly doses of cisplatin, 12 of these patients displayed a recurrence. Moreover, a dose reduction of radiotherapy was observed in 3 patients. None of these displayed a recurrence.”

  1. Why was HNSCC site not included as covariate? Prognosis of stage I laryngeal cancer is completely different from stage I hypopharyngeal cancer. How was the distribution of recurrences among the different HNSCC sites? I estimate hardly any recurrences in the HPV-positive OPSCC group, which is 43% of your study population.

Answer: This is true, and the reason we did not include site was that the exploratory data analysis indicated that this covariate was not important in our dataset. Since we tracked 9 different site categories, including site in the Cox regression would unnecessarily increase the dimensionality and also the variability. We have now included a variable selection step where we used the powerful multidimensional and non-linear Boruta variable selection algorithm. We included HNSCC site and treatment as covariates as a model validation step, and the Boruta method conclusively measured no importance of this variable in our dataset, meaning that any association is no stronger than random noise. This data can be found in Figure S1.

  1. Why was treatment not included as covariate?

Answer: Please see the answer to Q6.

  1. Why was smoking status only registered at diagnosis? Was there any information on whether patients continued smoking during/after treatment or if they stopped? It is known that continuing smoking during radiotherapy significantly decreases treatment effect due to tissue hypoxia from smoking (Chen et al Int J Radiat Oncol Biol Phys 2011).

Answer: This is an interesting question, but we unfortunately do not have the data to follow up this association.

  1. From the 272 patients included, 117 (43%!) are HPV-positive OPSCC. HPV is known to be the strongest prognostic factor in HNSCC, yet you decided not to include them as covariate “due to the high number of patients with p16/HPV positive oropharyngeal cancer and lack of data in other tumor locations”. I do not understand this. It is generally known that HPV-status does not affect prognosis in other HNSCC sites, so you could have made two groups (HPV-positive OPSCC versus HPV-negative HNSCC). Or you could have excluded HPV-positive OPSCC at all. But you cannot exclude them as covariate when it’s prognostic impact is so strong.

Answer: We agree with the reviewer, so far there is no available knowledge that HPV-status (or p16 status) in other HNSCC affects the prognosis. However, in the present study we aimed not to focus on HPV status as we have only p16/HPV status for patients with oropharyngeal cancer and not for the other groups of patients. Including the limited information we have would be skewed since we do not know the HPV status of the remaining patient population. Unfortunately, the HPV+- patients on their own are too few for a reliable statistical analysis.

  1. In Table 2 it is stated that BMI was on average 26.7 at baseline and 25.5 post-treatment, yet the BMI change posttreatment is zero, how is that possible?

Answer: We thank the reviewer for thoroughly reading our tables, but in this case it is correct as is. Since we are measuring BMI change, we measure from baseline (post treatment follow up), i.e. a landmarking approach. The other factors are instead measured as absolute CRP etc. and adjusted in the Cox model by their value pre-treatment. This means that the BMI change by definition is 0 post treatment. We have added clarifying information in the methods and materials regarding this.

  1. Did you check whether post-treatment CRP was correlated to therapy? The post-treatment timepoint was set 7 weeks after start of treatment, but I would expect that in patients that only required surgery, CRP is lower 7 weeks postoperative than in patients requiring a radiotherapy regimen of 7 weeks, in which the last RT fraction was given right before the post-treatment timepoint.

Answer: This is an interesting topic, and univariate analysis using one-way Anova does indeed show a significantly lower CRP expression in patients only receiving surgery. Between the other treatment groups there is no correlation. This is in line with previous findings from our group that surgery has little effect on protein markers for inflammation, while chemo/chemoradiation does have a large effect. In this manuscript however, surgery alone was mainly used in patients with Stage 1-II at admission, where recurrence rates were low. This codependency of CRP and treatment does not affect our conclusions. We have added to the discussion, line 538. “One possible uncertainty is CRP expression depending on other covariates in the study, for example stage or treatment type. For example, a recent study by Astradsson et al. demonstrated differential expression patterns of inflammatory proteins and immune response markers between patients that underwent surgery, radiotherapy and/or chemoradiation. This study showed that cisplatin-based chemoradiotherapy had im-munological effects in HNSCC patients (45), whereas surgery alone did not. In our current study, we observed a significantly lower CRP level at the post-treatment follow-up in patients receiving surgery as only treatment. There was no correlation between the other treatment groups as assessed by univariate ANOVA analysis (data not shown). Since patients receiving surgery only are over-represented in early clinical stages in our data, and stage was strongly correlated with recurrence, there is a risk of confounding in the connection between CRP and recurrence.”

  1. You state that “patients with elevated pre- and post-treatment CRP levels are at higher risk for recurrence of disease”. But did you check why these patients had elevated CRP levels? Did they have comorbidities? Active infections? Immune-suppressant medications? Any contra-indications to receive optimal treatment? Were they fit enough to finish the whole treatment regimen?

Answer: We have only included patients with a WHO performance status 0-2. Patients were considered fit to finish the whole treatment regimen by a multidisciplinary board. We have no data that can guarantee reliable information on comorbidities on all patients, which makes analysis difficult in this dataset. Immunosuppressive treatment was an exclusion criterion, this has now been added to the materials and methods, line 157-158. “Exclusion criteria included prior treatment of malignant tumors within the last 5 years (with the exception of skin cancer), immune suppressant treatment, severe alcohol problems, cognitive impairments, or other inability to participate or inability to understand Swedish.”

  1. And while you state that elevated CRP is associated with higher risk for recurrence, later on in the paper it seems to be true only for residual disease, and not for loco-regional recurrence.

Answer: We thank the reviewer for this insight, and it is true, the association is strongest for residual disease. Likely there are too few cases in the other subgroups, that is why we get less significance. This is the same reason why subgroup analysis of stage does not yield significant data, even though the association is very strong to each stage in the full and reduced Cox model.

  1. You state that “BMI post-treatment was strongly correlated with recurrence”, but are patients with a low BMI not also at increased risk of not being able to complete their treatment? Did you correct for this? Did these patients follow-up on the advices the dietician gave them? I think the Cox proportional hazards model currently includes too few covariates to make this statement.

Answer: We thank the reviewer for this suggestion, but we have not followed up patient compliance to the dietist and nutritional program in this dataset. Thus, we cannot correct for this. We have added the information about nr. of non-completed treatments (Please see Q5), and the number is very low (39). This makes subgroup analysis with regard to BMI for these patients difficult.

Round 2

Reviewer 2 Report

I would like to thank the authors for incorporating the feedback provided by the reviewers so quickly and thoroughly. It improved the overall quality of the manuscript. However, I do have one remaining concern:

1. I understand that HPV-status cannot be included as covariate in the analyses since you only have HPV-status for OPSCC and not for the other HNSCC sites. However, HPV-status is such a significant prognosticator in HNSCC that you cannot simply ignore it in my opinion. Is it an option to include it in HNSCC site? So that you have HPV-positive OPSCC, HPV-negative OPSCC, oral cavity SCC, hypopharyngeal SCC, laryngeal SCC, etc.